# In-orbit operation of an atomic clock based on laser-cooled $^{87}$Rb atoms

Liang Liu[1], De-Sheng Lü[1], Wei-Biao Chen[2], Tang Li[1], Qiu-Zhi Qu[1], Bin Wang[1], Lin Li[1], Wei Ren[1], Zuo-Ren Dong[2], Jian-Bo Zhao[1], Wen-Bing Xia[2], Xin Zhao[1], Jing-Wei Ji[1], Mei-Feng Ye[1], Yan-Guang Sun[2], Yuan-Yuan Yao[1], Dan Song[1], Zhao-Gang Liang[1], Shan-Jiang Hu[2], Dun-He Yu[2], Xia Hou[2], Wei Shi[2], Hua-Guo Zang[2], Jing-Feng Xiang [1], Xiang-Kai Peng[1] & Yu-Zhu Wang[1]

Atomic clocks based on laser-cooled atoms are widely used as primary frequency standards. Deploying such cold atom clocks (CACs) in space is foreseen to have many applications. Here we present tests of a CAC operating in space. In orbital microgravity, the atoms are cooled, trapped, launched, and finally detected after being interrogated by a microwave field using the Ramsey method. Perturbing influences from the orbital environment on the atoms such as varying magnetic fields and the passage of the spacecraft through Earth's radiation belt are also controlled and mitigated. With appropriate parameters settings, closed-loop locking of the CAC is realized in orbit and an estimated short-term frequency stability close to $3.0 \times 10^{-13} \tau^{-1/2}$ has been attained. The demonstration of the long-term operation of cold atom clock in orbit opens possibility on the applications of space-based cold atom sensors.

---

[1] Key Laboratory of Quantum Optics, Shanghai Institute of Optics and Fine Mechanics, Chinese Academy of Sciences, Shanghai 201800, China. [2] Research Center of Space Laser Information Technology, Shanghai Institute of Optics and Fine Mechanics, Chinese Academy of Sciences, Shanghai 201800, China. Correspondence and requests for materials should be addressed to L.L. (email: liang.liu@siom.ac.cn) or to D.-S.Lü. (email: dslv@siom.ac.cn) or to W.-B.C. (email: wbchen@siom.ac.cn)

Modern time keeping systems (TKS) on Earth and the global navigation satellite system (GNSS) rely heavily on atomic clocks. Traditional atomic clocks which use hot atoms, however, have almost reached their limits especially in regard to long-term stability. Laser cooling of atoms provides an approach to improve the performance of atomic clocks further[1], particularly in applications that require precision time-keeping over long time scales. The atoms are first cooled by lasers, and then interrogated by a microwave field typically with the Ramsey method. The width of the central Ramsey fringe for a cold atom clock (CAC) is almost two orders of magnitude narrower than that for their hot atom counterparts. A variety of CACs have been demonstrated on the ground, notably atomic fountain clocks[2–4] and optical frequency standards based on neutral atoms in a lattice or trapped ions[5,6]. Primary caesium fountain standards currently reach an uncertainty around $2 \times 10^{-16}$, and the improved accuracy and stability of optical clocks motivates a future redefinition of the SI second[7].

Currently, the best performing space atomic clocks used in the GNSS are those at a frequency stability of a few parts in $10^{15}$ per day[8]. Applying CACs in space is of great interest, not only in constructing the next-generation TKS and GNSS, but also in permitting deep space surveys and conducting more accurate tests of fundamental physics[9–13]. Moreover, other space applications in cold atom physics such as cold atom interferometry, optical clocks, and cold atom sensors also benefit from the techniques used in space CACs[14].

Experiments related to cold atoms in microgravity have been successfully demonstrated in a drop tower, parabolic flights, and a sounding rocket[15–18]. These methods provide a microgravity environment ranging from several seconds (drop tower, parabolic flight) to several minutes (sounding rocket). Nevertheless, testing while in orbital operation is required to gauge the long-term operation of a space CAC. Several projects on space CACs, such as ACES, PARCS, and RACE, have been proposed in the last few decades[19]. For example, the ACES mission, which consists of a caesium CAC called PHARAO, a hydrogen maser, as well as a package for frequency comparisons and distribution, aims to search for drifts in fundamental constants and measure the gravitational red shift with improved precision[20–26]. The PHARAO clock is expected to operate in space with a frequency stability of $1.0 \times 10^{-13}\tau^{-1/2}$ ($\tau$ is the average time in second) and an accuracy below $3 \times 10^{-16}$ (ref. [24]). Under the support of the China Manned Space Program (CMSP), we started a mission called Cold Atom Clock Experiment in Space (CACES) in 2011 with the goal of operating a rubidium CAC in space.

Operating a CAC in orbit has great challenges. First, because of the limited resources on board a spacecraft, weight, volume, and power consumption must be greatly reduced compared with ground-based fountain clocks. Second, the CAC must pass mechanical, thermal, and electromagnetic compatibility tests specified for space missions. Third, all operations of the CAC must be automated and all units must be maintained without any manual-adjustments. Fourth, the CAC must be robust in the orbital environment against, for example, the variation in Earth's magnetic field and impacts from high-energy particles. Finally, the CAC must be designed to work in microgravity.

## Results

**Experimental setup**. The principle of our Rubidium ($^{87}$Rb) space CAC is shown in Fig. 1. Because $^{87}$Rb has a smaller collision shift than that of $^{133}$Cs, better long-term performance can be expected with a relaxed control required over the atomic density[27,28]. The atoms are cooled and trapped in a magneto-optical trap (MOT); the cooled atoms are then launched using the moving molasses technique. Differing from that on the ground, cold atoms launched in microgravity move in a straight line at constant velocity. After state selection, the cold atoms are interrogated by a microwave field and their state is detected through laser-excited atomic fluorescence.

The whole system consists of four units: physics package, optical bench[29], microwave source, and control electronics. The main part of the physics package is a titanium alloy vacuum tube for which a vacuum is maintained to better than $1 \times 10^{-7}$ Pa[30]. A ring cavity inside the vacuum is used for the Ramsey interrogation of the cold atoms[31]. Three layers of Mumetal are used to shield the magnetic field in the interrogation cavity, with only one layer shielding the capture zone. The magnetic field in the interrogation region is stabilized to better than 5 nT using a servo loop and a magnetic field coil, which compensate for field variations arising from the motion of the spacecraft in orbit[32].

The MOT is formed using a pair of anti-Helmholtz coils and two laser beams with independently controlled frequencies. Each beam is folded to create the multiple trapping beams required[33]. The intensity of each trapping beam is about 3.8 mW cm$^{-2}$ and the geometry is designed so that a frequency shift between the beams forms a moving molasses that launches atoms towards the interrogation region with a velocity

$$v = (\omega_1 - \omega_2)/k, \qquad (1)$$

where $\omega_1$ and $\omega_2$ are the frequencies of the two laser beams, and $K$ is the wave vector. Launch velocities ranging from 0.6 to 6.0 m s$^{-1}$ is accessible. With this process, about $5 \times 10^7$ cold atoms are launched from the MOT zone when the loading time is set to 1.0 s.

State selection is accomplished with a combination of microwave excitation and laser pushing. After being cooled, the

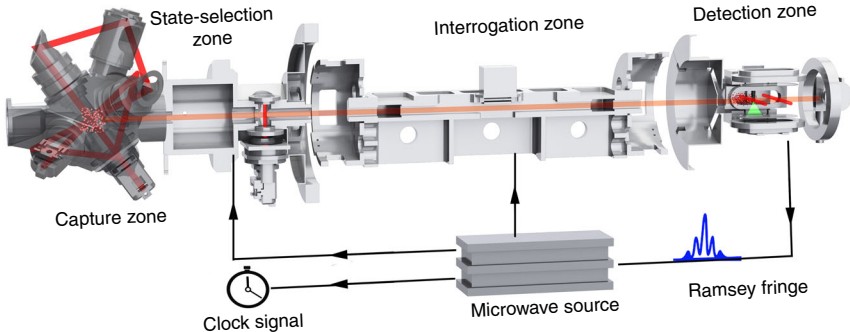

**Fig. 1** Principle and structure of the space cold atom clock (CAC). The capture zone is a magneto-optical trap (MOT) with a folded beam design. The ring interrogation cavity is used for the microwave field to interrogate the cold atoms. In the detection zone, cold atoms in both hyperfine states are detected. The clock signal is obtained by feeding the error signal to the frequency of microwave source

atoms are concentrated in the state $|F = 2\rangle$ and evenly distributed in the five magnetic sub-states. The microwave power in the state selection cavity is adjusted such that the atoms in $|F = 2, m_F = 0\rangle$ are pumped to $|F = 1, m_F = 0\rangle$ at an efficiency of almost 100%; the laser beam then pushes away all other atoms in the states $|F = 2, m_F \neq 0\rangle$. The remaining atoms are used for microwave interrogation.

In two subsequent interactions of duration $\Delta t$ with the interrogating field separated by a free evolution time $T$, the Ramsey interrogation of the space CAC employs two cavities separated by a distance $D$, giving an interrogation time $T = D/v$ for cold atoms moving with velocity $v$ in the microgravity of space. The full-width-at-half-maximum (FWHM) $\Delta$ of central Ramsey fringe is directly related to the velocity[34]

$$\Delta = v/2D \ (T \gg \Delta t, \Delta\omega \ll \Omega), \qquad (2)$$

where $\Delta\omega$ is the microwave frequency detuning from resonance, and $\Omega$ the Rabi frequency. Taking into consideration the dead time of the clock cycle and the size limitation, $D = 217$ mm is used in our space CAC.

Our space CAC has been tested in the laboratory and all of its performance metrics meet the design specifications[35]. The setup passed all mechanical, thermal, and electromagnetic compatibility tests required by CMSP. It was launched into orbit on 15 September 2016 with the Chinese space laboratory Tiangong-2 and put into operation the following day. Since then, the space CAC has been working in orbit under the management of Tiangong-2. Under almost continuous operation in microgravity, our space CAC has already been tested for over 15 months in orbit.

**Clock operation.** Figure 2 presents typical results of time-of-flight (TOF) signals in the detection region. The cold atoms are launched by the moving molasses technique at the specified velocity Eq. (1), and detected by laser-excited atomic fluorescence. The transition probability $p$ is found from

$$p = \frac{N_2}{N_1 + N_2}, \qquad (3)$$

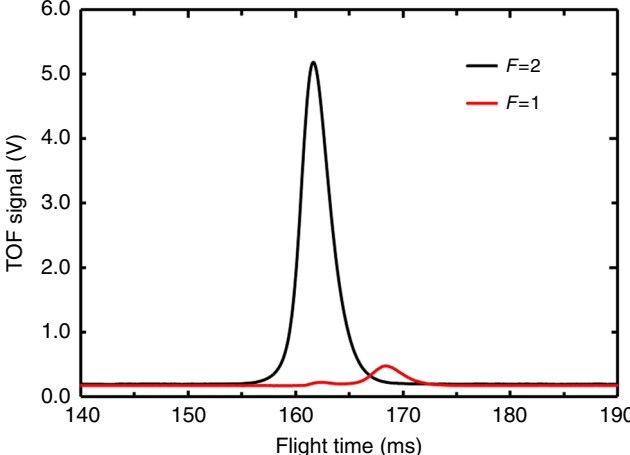

**Fig. 2** Typical time-of-flight (TOF) signal. Flight time begins when the cold atomic cloud is launched. The red and black curves correspond to the fluorescence from states $|F = 1, m_F = 0\rangle$ and $|F = 2, m_F = 0\rangle$, respectively. The launch velocity is 4 m s$^{-1}$

where $N_1$ and $N_2$ denote the number of cold atoms in the states $|F = 1\rangle$ and $|F = 2\rangle$, respectively. With this normalized detection, influences from fluctuations in atomic number on clock measurements are greatly reduced.

Figure 3a–d presents typical central Ramsey fringes of the space CAC. The interrogation cavity and microwave frequency are pre-tuned to the atomic transition frequency between the two ground states $|F = 1, m_F = 0\rangle$ and $|F = 2, m_F = 0\rangle$. The injected microwave power is optimized by measuring the transition probability $p$, Eq. (3). When the power is such that each passage through the microwave cavity applies a $\pi/2$ pulse all of the population flips from $|F = 1, m_F = 0\rangle$ to $|F = 2, m_F = 0\rangle$ on resonance. Unlike atomic fountains under gravity on the ground, the FWHM of the central Ramsey fringes in microgravity, Eq. (2), is linearly related to the launch velocity, as shown in Fig. 3e.

To determine the parameter settings for the clock operation, we measured the signal-to-noise ratio (SNR) for different launch velocities. Even though the lower velocity gives a narrower FWHM (Fig. 3e), it also leads to a lower SNR mainly due to the loss of cold atoms originated from expansion of cold atom cloud. Combining the contributions of both SNR and FWHM, a 1.1 m s$^{-1}$ launch velocity, which corresponds to an FWHM of 2.0 Hz, yields an optimal performance. At this velocity, an SNR of 440 at the half-maximum point of the central Ramsey fringes is achieved (Fig. 4b). Figure 4a presents a plot of the population oscillation vs. microwave power at resonance frequency to determine the power required for $\pi/2$ transition in each interaction zone. The calculated results agree well with the measured data except the oscillations at higher microwave power mainly due to the velocity distribution of the cold atomic cloud and the inhomogeneities in the microwave amplitude. We focused on the first peak, from which we got the microwave power of about −66.5 dBm for the required $\pi/2$ transitions; the corresponding Ramsey fringes are given in Fig. 4b.

Using the Ramsey fringes with 2.0 Hz FWHM, a closed-loop operation has been performed on the CAC by feeding the error signal to the direct digital synthesizer (DDS) of the microwave source. Figure 5 gives the error signal for the transient stage of the closed-loop operation. After a transient time of about 300 s, the error signal is stabilized around zero, which means the microwave frequency is tightly locked to the atomic resonance. This operation has been continuously performed for around 1 month before intentional interruption.

To maintain continuous clock operation for long periods in orbit, several issues distinct from conditions typical on the ground are of concern. First, the environmental magnetic field varies periodically by about 80 μT due to the motion of the spacecraft in low Earth orbit (LEO). Figure 6 shows the variation of the magnetic field inside the outer magnetic shield monitored by a fluxgate magnetometer mounted around the state-selection cavity. The roughly 90 min period corresponds to the orbital period of the spacecraft around Earth. If left uncontrolled, this varying magnetic field would degrade the magnetic uniformity inside the interrogation zone and leads to clock frequency variations arising from the Zeeman effect. In the CAC, a magnetic field compensation loop is used and the parameter settings of the loop has been specified by the detected magnetic field using the fluxgate in orbit. The magnetic field along the trajectory of the cold atoms inside the interrogation zone is measured by exciting the magnetic-sensitive transition of rubidium[24] and a variation of 4.5 nT is obtained when the loop is active. Such fluctuations lead to a clock frequency instability of about $1.7 \times 10^{-16}$ on the timescale of the field fluctuations.

Second, in LEO altitudes, an area of the Van Allen radiation belt known as the South Atlantic Anomaly (SAA)[36] is observed in our test in orbit (Fig. 6). A spacecraft in LEO passing through that

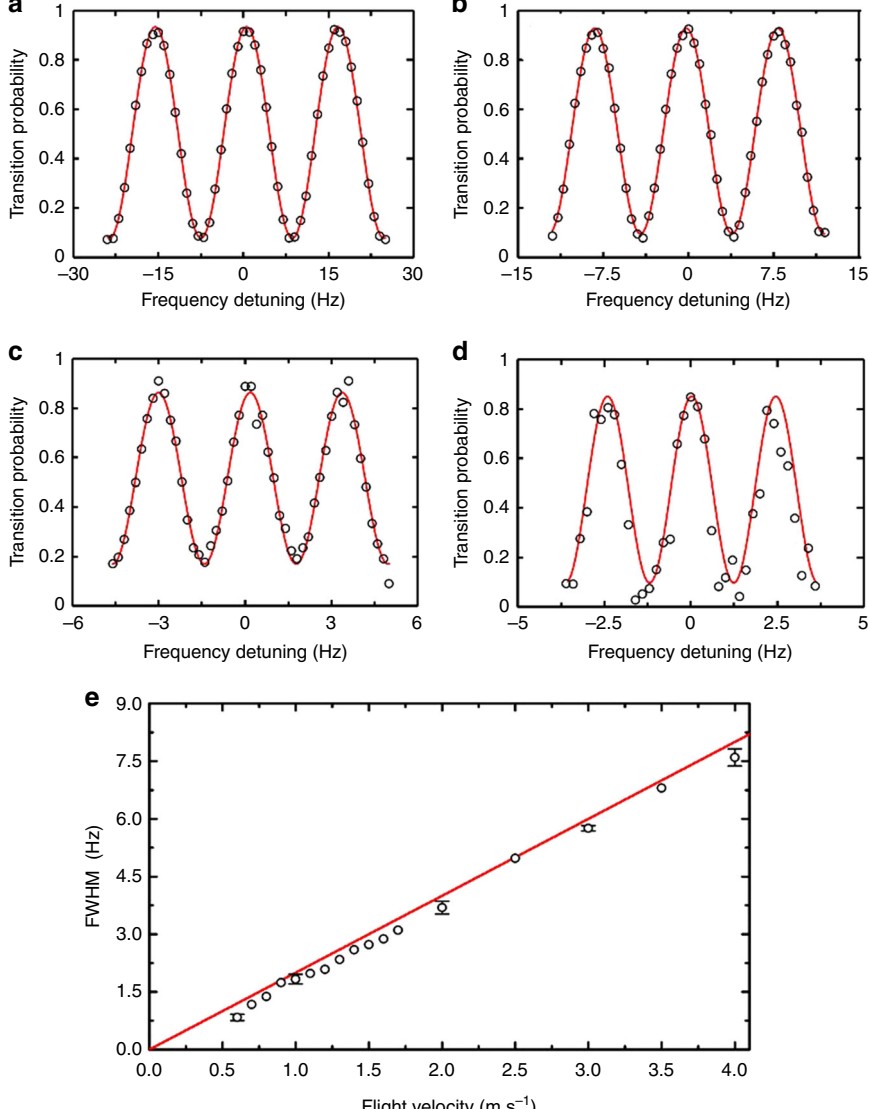

**Fig. 3** Ramsey fringes with different launch velocities. Central Ramsey fringes of the space CAC with a launch velocity of **a** 4.0 m s$^{-1}$, **b** 1.0 m s$^{-1}$, **c** 0.8 m s$^{-1}$, and **d** 0.6 m s$^{-1}$, respectively, corresponding to the FWHM of central fringe 7.3, 1.8, 1.4, and 0.9 Hz. Red lines are sinusoidal fits. **e** Dependence of FWHM of the central Ramsey fringes on flight velocity. Red line represents calculated result; black circles represent measured results. Each error bar is the standard deviation calculated from eight measurements

region is exposed to high-energy particles. Long-term observations indicate that this radiation has no significant influence on laser cooling. However, it does interfere with the photodetectors inside the detection zone and introduces random spikes on the detected TOF signals. There are two interference issues: spikes located on the TOF signal and near the TOF signal (see Fig. 7). When spikes are actually on the TOF signal (Fig. 7a), the data are discarded. When spikes are near the TOF signal but well separated from the signal (Fig. 7b), a window function is used to filter out the TOF signal from the contaminated signal. All these processes are automatically managed by the software of the control units. Because the issue in Fig. 7b is the majority, the influence from discarded data (about 0.05% of the total data) to the clock operation is slight and can be ignored.

Third, in most cases during Tiangong-2's flight, the microgravity level is stationery with fluctuations of about a few parts in $10^4$ g, which does not make a difference to the clock operation. However, when the spacecraft changes attitude, the microgravity levels inside vary over a large range. In this instance, the clock

data are invalid and automatically discarded by the software. Nonetheless, adjustments in spacecraft attitude are seldom and predictable in flight, and therefore its influence on the clock operation is controllable. Apart from these three influences, other impacts such as fluctuations in ambient temperature and atmosphere pressure are also well managed.

As mentioned above, a longer interrogation time leads to a narrower linewidth but also more loss of cold atoms, which reduces the SNR of the Ramsey fringes. Using the SNR of 440 and FWHM of 2.0 Hz, a short-term frequency stability at an averaging time $\tau$ is close to $3.0 \times 10^{-13} \tau^{-1/2}$ as expected with a clock period of 2.0 s[37]. There is no second clock on board and also no frequency dissemination link to the ground, so we have no reference with which to evaluate the clock stability in orbit. However, we have verified the performance of the CAC in our laboratory by measuring the frequency stability against an H maser as shown in Fig. 8. The calculated results estimated from the measured SNR and FWHM of the central Ramsey fringes are compared with the ground-based measurements (Fig. 8). The

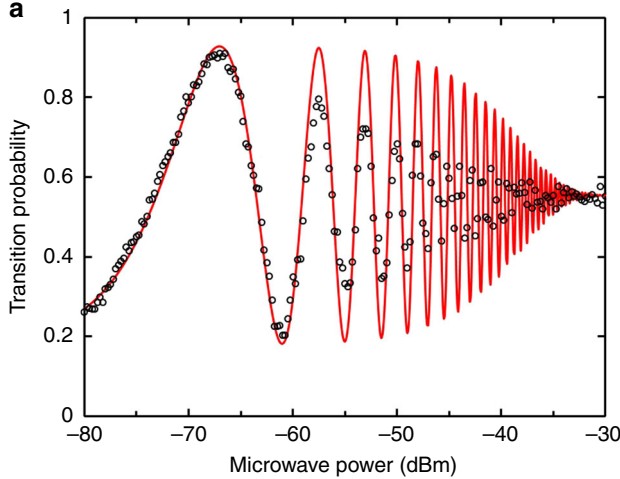

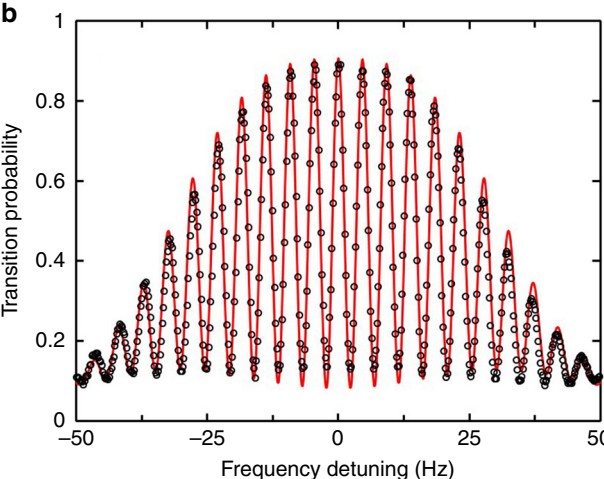

**Fig. 4** Atomic population oscillations after microwave interrogation. **a** Population oscillations vs. microwave power and **b** Ramsey fringes in microgravity with a launch velocity of 1.1 m s$^{-1}$, used in closed-loop operation. The black circles represent measured data and the red lines are calculated results

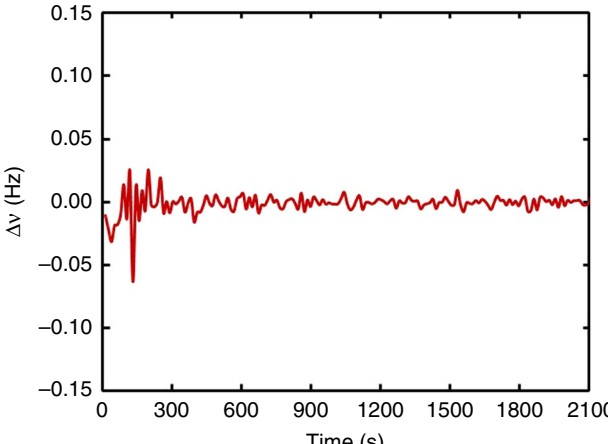

**Fig. 5** The error signal fed to the frequency of microwave source. Deviation of the microwave frequency from the atomic transition after locking the clock signal to the DDS of the microwave source. The servo loop is activated at time $t = 0$

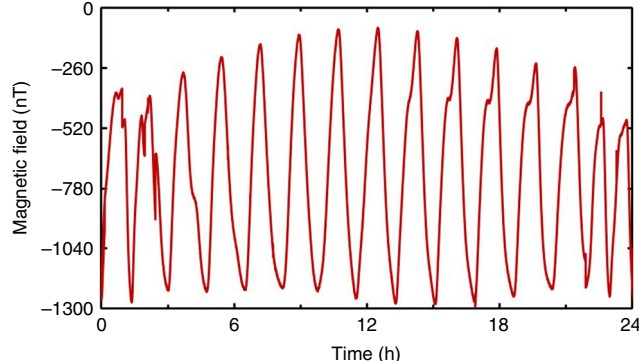

**Fig. 6** Magnetic field inside the outer magnetic shield vs time as the spacecraft rotates in LEO. Every peak corresponds to one orbit and the dips shown in some peaks correspond to the magnetic field when spacecraft moves through the South Atlantic Anomaly (SAA) region

estimated instability in orbit is more than a factor of six lower than on Earth, which we attribute to the microgravity environment. Such measurements on the ground hint at the long-term performance of the space CAC. Currently, the frequency stability of the space CAC is mainly limited by the detected atoms. This limitation will be improved in the next space CAC for the Chinese space station by increasing the number of cold atoms using a 2D MOT and larger power for the cooling lasers. As the collision shift is low, more $^{87}$Rb atoms can be used without degrading the accuracy of the CAC.

We realized laser cooling of $^{87}$Rb atoms and achieved the launching of cold atoms at low velocity in microgravity, demonstrated long-term closed-loop operation, and estimated the performance of the CAC in orbit. Since launch, our space CAC has been working in orbit for more than 15 months, and its performance continues as designed. We anticipate our demonstration to be a starting point for more tests of space-based cold atom sensors, for example, the construction of the next-generation TKS and GNSS in deep space, as well as of optical clocks, atomic interferometers, atomic gyros, etc. Furthermore, the experience from this mission, especially the data related to the orbital environments, will also benefit precision physics experiments such as the preparation of the ultra-cold quantum gases.

## Methods

**Setup of the space CAC**. The space CAC (Supplementary Figure 1) consists of four sub-systems: physics package, optical bench, microwave source, and control electronics.

The physics package is an ultra-high vacuum (UHV) tube surrounded by a three-layer magnetic shield in which the rubidium atoms are cooled, state-prepared, interrogated by the microwave field, and detected. The UHV tube is closed by a rubidium base on one side and by two 21 s$^{-1}$ ion pumps on the other side. The rubidium source is covered by a thin film with tens of 0.1 mm diameter holes to avoid the leakage of liquid rubidium in microgravity. The atom flux is regulated by the temperature of the rubidium base. The rubidium vapour diffusing into the capture zone is captured and cooled by a MOT. Unlike the traditional configuration with six laser beams, the MOT we use is compact with only two input laser beams employing a fold-optical-path method. Its optical path and a photograph are shown in Supplementary Figure 2. Following the capture zone, a TE$_{011}$ cylindrical microwave cavity with cut-off aperture of 13 mm diameter is used as a state-selection cavity. To avoid rubidium migration from the capture region to the other zones, a graphite getter is inserted in the tube between the capture zone and the state-selection cavity.

The interrogation cavity of the CAC is designed as a rectangular waveguide cavity (Supplementary Figure 3), based on the U-type interrogation cavity. The microwave signal propagates symmetrically along the two guided wave zones (TE$_{107}$ mode) and forms a standing wave field at each microwave interaction zone (TE$_{201}$ mode). Therefore, the one-way-flight cold atoms interact with the interrogating microwave field twice along their trajectory. This cavity is made of titanium alloy coated with silver to a thickness of roughly 5 μm. The measured loaded quality factor is approximately 4200. To mitigate the influences of the fluctuations in environmental temperature on the cavity resonance frequency, an

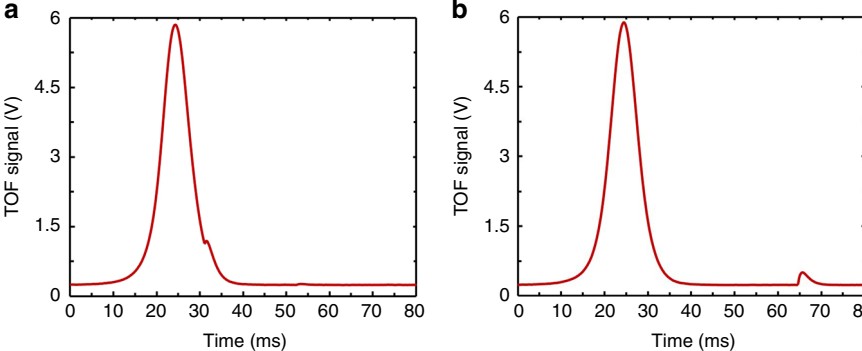

**Fig. 7** Interfered TOF signals in the SAA region. Interference spikes located **a** on the TOF signal and **b** near the TOF signal when the spacecraft moves in the SAA region. Time $t = 0$ is the start time of the detection process

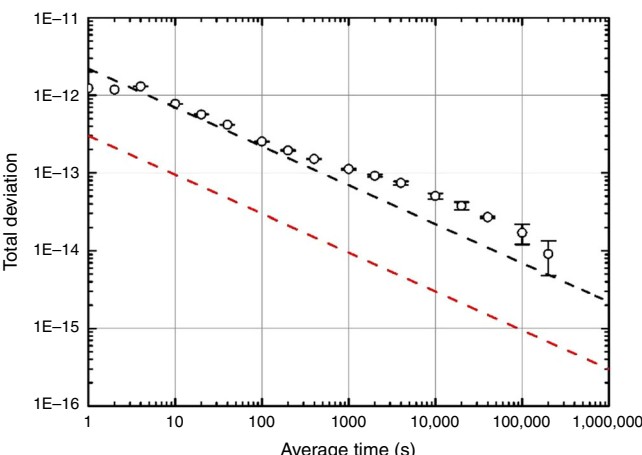

**Fig. 8** Frequency stability of the space clock. Black circles represent the measured total deviation against an H Maser on the ground; the black dashed line represents the predicted frequency stability calculated using the measured SNR and FWHM on the ground. The cold atomic cloud is launched downward with a velocity of 1 m s$^{-1}$ on the ground. The red dashed line represents the predicted frequency stability in orbit. The error bars represent uncertainty in the total deviation estimator

active thermal controller is used to stabilize the cavity temperature to about 25 °C with a stability of ±0.01 °C.

Finally, the cold atoms enter the detection zone and pass through four laser beams in succession. The laser beams are set to perpendicular to the trajectory. The first beam is circularly polarized and forms a standing wave tuned to the transition $\left|5^2S_{1/2}, F = 2\right\rangle \rightarrow \left|5^2P_{3/2}, F = 3\right\rangle$ to detect the population in state $\left|5^2S_{1/2}, F = 2\right\rangle$. Tuned to the same transition as the first, the second beam is a travelling wave to push away the atoms in state $\left|5^2S_{1/2}, F = 2\right\rangle$ after their detection. The third beam is tuned to the transition $\left|5^2S_{1/2}, F = 1\right\rangle \rightarrow \left|5^2P_{3/2}, F = 2\right\rangle$ and pumps the atoms in state $\left|5^2S_{1/2}, F = 1\right\rangle$ to state $\left|5^2S_{1/2}, F = 2\right\rangle$. Finally, the fourth beam is set up like the first but detects the remaining atoms. As designed, the detection system (Supplementary Figure 4) is a compact integrated opto-mechanical system. All the optical components are glued, to improve the thermostability of the detection system, which is athermalized. The induced fluorescence is recorded by two 10 mm ×mm photodiodes with an efficiency of 2% and the photocurrent of each photodiode is converted into voltage using amplifiers with a gain of 10$^7$ A V$^{-1}$. The photodetection has a noise level of less than 3 µV Hz$^{-1/2}$ over the frequencies from 1 to 100 Hz which can be ignored in the experiment. The two photodetection signals are digitized by the control electronics to calculate the transition probability.

The vacuum tube is built with four individual chambers, each is connected by flanges to the others. Each flange is sealed by Delta seals and indium whereas the optical windows are sealed by indium. To improve the vacuum level of the interrogation zone, two groups of four getters are distributed on the internal sides

of the vacuum chamber at the two ends of the interrogation zone. All these methods allow the tube to maintain an UHV for a long time even if the ion pumps do not work. The vacuum level of the CAC in orbit is less than $1 \times 10^{-7}$ Pa.

The vacuum tube is surrounded by three cylindrical Mumetal shields (Supplementary Figure 5): inner shield, middle shield, and outer shield. The total magnetic attenuation of these three-layer shields is more than 36 dB and the variation of the magnetic field inside the shield along the atom trajectory is less than 2 nT in a static geomagnetic field (Supplementary Figure 6). The interrogation zone is inside the inner shield, a solenoid coil surrounds the interrogation cavity to provide a clock magnetic field of about 100 nT. The middle shield encloses the vacuum tube from the state-selection cavity to the detection zone. One compensation coil is mounted close to the inner magnetic shield aperture to improve the magnetic homogeneity; a second coil is wound around the state-selection cavity to define the magnetic field for state selection. The outer shield encloses the whole tube and a coil is mounted on the capture zone. Two magnetic fluxgates are placed inside the magnetic shield. The magnetic field inside the middle shield can be detected by the fluxgate mounted near the detection zone and used as feedback to the current in the interrogation coil to compensate the fluctuation of the external magnetic field. The fluxgate mounted near the state-selection cavity is used for monitoring.

The optical bench is designed as a compact, robust optical system. Its architecture is shown in Supplementary Figure 7. The laser sources we use are distributed Bragg reflector (DBR) diodes. Each laser diode (LD) is followed by a 40 dB isolator to eliminate feedback from the surfaces of the optical components. The frequency of cooling LD1 is locked to the saturated absorption crossover resonance of the transition $\left|5^2S_{1/2}, F = 2\right\rangle \rightarrow \left|5^2P_{3/2}, F = 2, 3\right\rangle$ of the $^{87}$Rb D$_2$ line through an acousto-optic modulator. The whole frequency stabilization process is automated and managed by a microcontroller unit. The laser beam from cooling LD1 is divided into four beams, which are coupled into different fibres and used as two cooling laser beams, the state selection laser beam, and the probing laser beam. The frequency of each beam is shifted by an acousto-optic modulator before the fibre coupler. To improve the reliability, LD2 is used as a backup laser source of the cooling LD1. The repumping laser, LD, is frequency-locked to the saturated absorption crossover resonance of the transition $\left|5^2S_{1/2}, F = 1\right\rangle \rightarrow \left|5^2P_{3/2}, F = 1, 2\right\rangle$ of the $^{87}$Rb D$_2$ line. The beam passes through an acousto-optic modulator and is split into two beams. One beam is coupled into a fibre and act as a repumping laser beam for the dual-level detection process, and another beam is combined with the cooling laser beam for the laser cooling process.

The base of the optical bench is made of SiC-reinforced Al–Si alloy, its size is 300 mm × 290 mm × 30 mm. All optical components were validated to comply with the space-application criteria, and their mounts were specially designed and manufactured. Additionally, the temperature of the optical bench is stabilized to about 25 °C, which is the optimum operating temperature of this system. Both active and passive methods are used for temperature stabilization. The active temperature control is used to compensate the thermal loss of the optical bench, whereas the passive temperature control is used to dissipate excess heat from the hot component and to maintain a maximum temperature gradient below 1.0 °C for the whole optical bench.

The microwave source is based on frequency multiplication of an ultra-low phase noise quartz oscillator (BVA8607, OSA) up to 6.834 GHz (Supplementary Figure 8). The mass is 5 kg and the power dissipation is 20 W. The BVA8607 oscillator output signal has excellent spectral purity but is fragile to mechanical vibration during the rocket's launching phase. To circumvent this problem, the oscillator is placed in an aluminium enclosure with polyurethane hemisphere vibration isolators attached to each inner sidewall. With this method, the BVA8607 oscillator has passed the space vibration qualification test.

The oscillator frequency (5 MHz) is multiplied to 100 MHz which is used to phase-lock a local 100 MHz quartz oscillator with a bandwidth of about 100 Hz. The purpose of this loop is to optimize the frequency noise spectrum of the

100 MHz signal at high Fourier frequencies. The 100 MHz signal is multiplied to 200 MHz and then split into two signals. One is used to drive a DDS and two frequency down-converter mixers; the other is multiplied to 7 GHz. The 7 GHz signal is mixed with a 7.034 GHz signal from a dielectric resonator oscillator (DRO) to produce a fractional frequency signal (about 34.68 MHz), which is compared with the output signal of the DDS to phase-lock the DRO. Finally, the output signal of the DRO is split into two signals and each signal is frequency down-converted to 6.834 GHz. One 6.834 GHz signal is fed to the state-selection cavity whereas the other is fed to the interrogation cavity. Each microwave signal can be switched off by more than 65 dB to avoid significant microwave leakage. In addition, the power of the microwave signals can be adjusted to a resolution of 0.25 dB by digital attenuators.

The phase noise spectral density of the 6.834 GHz signal is mainly dominated by that of the 5 MHz oscillator. The contribution of other elements is 10 dB lower. Using the measured spectrum (Supplementary Figure 9), the frequency stability limited by the microwave source[4] is estimated to be $1 \times 10^{-13} \tau^{-1/2}$ which is consistent with the mission requirements.

The control electronics consists of two parts: the microcontroller unit and the FPGA unit. The microcontroller unit manages the data communication and calculations whereas the FPGA unit controls the operation of the CAC. The microcontroller unit is connected to the FPGA unit by a RS422 line for data exchanges. The FPGA unit sends trigger signals to other sub-systems at different phases of the clock cycle, for instance the AOM frequency changes, the microwave source frequency or the microwave switch operations. The FPGA unit also receives digitized analogue signals, the main signal being the detection fluorescence signal, and implements servo loops such as temperature control and magnetic field compensation.

**Experimental method in orbit**. The timing sequence of the laser trapping and cooling is presented in Supplementary Figure 10. This process is composed of four stages: capture, pre-cooling, launching, and post-cooling. The power and frequency of the cooling beams are regulated by the AOMs. First, the rubidium atoms are captured in the MOT. After hundreds of milliseconds, the atoms are cooled to about 100 μK and the magnetic field of the MOT is switched off. A few milliseconds later, the cooling process changes to a pre-cooling stage that lasts from 4 to 11 ms with different launching velocities where the atoms are further cooled by polarization gradient cooling. Thereafter, the cold atoms are launched with the moving optical molasses. During the launching stage, the atoms are slightly heated. To cool the atoms in the moving frame, an adiabatic cooling by slowly ramping down the intensity of the cooling beams together with the polarization gradient cooling are used in the post-cooling stage. Finally, the atoms are cooled to several micro-kelvins at the end of the cooling phase.

If the cold atom cloud has small density, its spatial distribution can be approximately considered to be Gaussian. In this experiment, Gaussian fitting is used to deduce the detected atoms number and their temperature. The number of the cold atoms is obtained from

$$N = \frac{\int U_{PD}(t)dt}{\gamma_P \cdot \Delta t \cdot c \cdot \rho_{PD} \cdot G \cdot E_{Rb}} \quad (2)$$

where $\gamma_P$, $\Delta t$, $c$, $U_{PD}(t)$, $\rho_{PD}$, and $G$ represent respectively the scattering rate of atoms, the time taken for atoms to cross the probing beam, the fluorescence collection efficiency of the detection system, the TOF signal lineshape in volts, the quantum efficiency of the photodiode and the gain of current amplifier. $E_{Rb}$ is the $D_2$ transition energy of $^{87}$Rb, which equals $2.54 \times 10^{-19}$J. In this experiment, we have $\gamma_p = 1.7 \times 10^7 s^{-1}$, $c = 2\%$, $\rho_{PD} = 0.5 AW^{-1}$, and $G = 1.1 \times 10_9$ V A$^{-1}$. By substituting these values into Eq. (2), the atomic number can be obtained.

From the ground-based tests, the initial size of the cold atom cloud is approximate 1 mm which is small compared with that of the detected cold atom cloud in the detection zone. From the principle of equipartition of energy, its temperature is found from

$$T = \frac{m}{k_B} \frac{\sigma_t^2 \cdot v_0^2}{\Delta t^2}, \quad (3)$$

where $m$ is the mass of a single rubidium atom, $K_B$ the Boltzmann constant, $v_0$ the velocity of the atomic cloud while passing through the probing beam, $\Delta t$ the time period between launch and detection, and $\sigma_t$ the Gaussian radius of the TOF signal, which represents the size of atomic cloud.

**Data availability**. The data that support the findings of this study are available from the corresponding authors upon request.

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

## Acknowledgements

We would like to thank our colleagues at the Technology and Engineering Center for Space Utilization, CAS, especially Yidong Gu, Min Gao, Guangheng Zhao, Congming Lü, Hongen Zhong, and many others for their many discussions and technical supports. We also thank Wenrui Hu of Institute of Mechanics, CAS, and Tianchu Li of National Institute of Metrology for their long-term discussions on the space applications of the atomic clock. We also thank Shenggang Liu and Yuanci Gao of the University of Electronic Science and Technology of China for his work on the microwave cavity, the Nanopa Vacuum Co. for their help on the vacuum tube. We thank K. Gibble, H. Metcalf, and Xinye Xu for their many suggestions on the manuscript. We acknowledge the supports from the China Manned Space Engineering Office, the Chinese Academy of Sciences, and the National Key R&D Program of China (Grant No. 2016YFA0301504).

## Author contributions

L. Liu and Y.-Z.W. conceived the research. L. Liu, D.-S.L., W.-B.C., T.L., Q.-Z.Q., B.W., L. Li, W.R., J.-B.Z., X.Z., J.-W.J., M.-F.Y., Y.-Y.Y., J.-F.X. and X.-K.P. designed the experiments. All authors designed and developed the setup. D.-S.L., J.-B.Z., W.R., and M.-F.Y. developed the physics package. W.-B.X., B.W., Z.-R.D., Y.-G.S., and Q.-Z.Q. developed the optical bench. T.L. and W.S. developed the microwave source. L. Li, X.Z., J.-W.J., S.-J.H., D.-H.Y., X.H., H.-G.Z., and Y.-Y.Y. developed the control electronics and software. W.R., J.-W.J., X.-K.P., J.-F.X., L. Liu, D.-S.L., T.L., Q.-Z.Q., B.W., and L.Li contributed to the data collection and discussed the results. D.S. and Z.-G.L. are responsible for project management. Y.-Z.W. supervised the whole project.

## Additional information

**Competing interests:** The authors declare no competing interests.

