## [Peer Review File · Nature Communications]

Reviewers' comments:

Reviewer #1 (Remarks to the Author):

The authors present the first cold-atom experiment conducted in space, which is an amazing technical achievement. There has been a lot of work to try to send an atomic clock based on laser-cooled atoms into space over the past 20 years. There are several motivations for doing this. For one, under microgravity the atoms can be interrogated for long periods. Another reason is that the high speeds and microgravity environment make the system well suited for precision tests of relativistic effects. A third motivation is that cold-atom systems can potentially improve on the performance of current spacebased atomic clocks for GNSS systems.

Efforts to launch cold-atom clocks into space include a program in the US called PARCS, which was ended after the 2003 space shuttle disaster for lack of a delivery vehicle to the ISS. There has also been the ACES project in Europe, which continues work to launch a cold-atom atomic clock into space, but which was eclipsed by this Chinese space clock presented here by Liang and colleagues. The work presented by Liu and colleagues is made even more impressive by the detail that the Chinese clock program presented here did not start until 2011 – over ten years later than the start of the ACES program.

Because of the huge impact of this work, I highly recommend that the work be published in Nature Communications. However, I have quite a few grammar corrections and suggestions to recommend before publication. I also have a few suggestions for clarifications that should improve the technical content of the paper as well. I list these corrections below.

Detailed suggestions:

1. I made several editing suggestions for the manuscripts grammar directly onto the submitted PDF. I recommend that the authors carefully consider all suggestions to improve the readability of the manuscript – especially to the general audience that will read the paper.
2. I would change the title to something like “In-orbit operation of an atomic clock based on laser-cooled ^{87}Rb atoms”
3. Line 16: The abstract emphasizes that the width of the fringes scales linearly with launch velocity, but I don't think this is such an important detail that it should be emphasized in the abstract. Beam clocks on the surface of the earth with the atom propagation direction perpendicular to gravity will also exhibit this behavior. I would remove this sentence from the abstract.
4. Line 67: I would clarify the statement about the reduced density shift in the Rb clock and why that can improve performance. In my opinion, the smaller shift allows for relaxed control of the Rb density. So instead of saying “better performance can be expected”, I would recommend saying something like “better long-term performance can be expected with relaxed control required over the atomic density.”
5. On line 74 and then throughout the paper, I would change “physical package” to “physics package”.
6. Equation 2 should be an equality not a proportionality.

7. The sentence starting on line 110 sounds simplistic and I would change it to say something like: “The continuous microgravity time in orbit has allowed us to test the space CAC over a period of XX months.”
8. The two sentences beginning on line 126 are strange. I would suggest changing them to something like this: “The injected microwave power is optimized by measuring the transition probability described in Eq. (3). When the power is such that each passage through the microwave cavity applies a $\pi/2$ pulse, all of the population flips from ...”
9. The sentence beginning on line 129 is not true for all clocks, for example beam clocks with the beam path orthogonal to gravity. So I would change the beginning of the sentence to say “Unlike atomic fountains under gravity on the ground, ...”
10. Do you know why there is a slight offset between the red line and the data points in Fig. 3? If so it would be good to comment about that bias.
11. The sentence beginning on line 143 needs clarification for a few reasons. Also, which cavity are you scanning the power in? The state selection or Ramsey cavity? If it’s the Ramsey cavity, don’t you want to apply $\pi/2$ pulses? And are you turning the microwaves off during the second passage through the cavity?
12. I’m not sure if I agree with the conclusion in the sentence beginning on line 144. At higher microwave power, you expect the contrast to go down because of inhomogeneities in the microwave amplitude across the atom cloud. I would argue that the reduced contrast is not from sparse measurements. But how is the theoretical curve calculated and what is assumed? It seems like there is another cause of decay that is not accounted for, or that the inhomogeneities in the cavity are larger than expected.
13. I think the discussion of figures 7A and 7B has the cases backwards in the text. Please check. Maybe you should change the text to say something like this: “When the spikes are actually on the TOF signal as shown in Fig. 7A, the data is thrown out. When the spikes are near the TOF signal but well separated from the signal, a window function is used to “filter out” the TOF signal from the contaminated signal.”
14. The discussion from lines 202 through the beginning of 206 is redundant with the earlier discussion of the SNR versus atom launch velocity.
15. I don’t think it will be clear to all readers why you can’t show an Allan Deviation for the clock performance in flight. I would suggest adding a sentence starting on line 208 to say something like this: “There is no second clock to compare the CAC to aboard the spacecraft, so we have no reference with which to evaluate the clock stability in orbit.”
16. The estimated stability in orbit is actually a lot better than what you showed on the ground. I would emphasize that by saying something like: “...the estimated instability in orbit is more than a factor of six lower than on earth, which we attribute to the microgravity environment...”
17. Caption of Fig. 8: Please specify what the launch velocity was for the ground based measurement and what the orientation of the CAC was with respect to gravity.

Reviewer #2 (Remarks to the Author):

The paper by Liang Liu et al. reports on the first demonstration of the operation of a cold atom clock in microgravity. Even if the reported results can be considered still as proof-of-principle, this is in my

opinion an important milestone towards experiments with cold atoms in space and therefore the paper is worth being published.

I would suggest however that the author consider the following comments in order to submit an improved version.

-> English should be checked and improved.

-> Since the work reported in this paper is similar to the ACES/Pharao CNES planned mission, ACES should be cited more clearly and the results of this work should be compared with the one expected from ACES.

-> Some references are missing and some could be omitted or replaced;

- Ref.s 3 and 4 could be omitted and replaced with relevant references on the characterization of the best fountain clocks.

- Reference to planned future experiments with atomic sensors in space should be included as, for example:

Atom interferometers and optical atomic clocks: New quantum sensors for fundamental physics experiments in space

Nuclear Physics B - Proceedings Suppl. Volume 166, Pages 159-165 (2007)

- References to ACES/Pharao should be included: a list from which the authors may choose is:

1. P. Laurent, P. Lemonde, E. Simon, G. Santarelli, A. Clairon, P. Petit, N. Dimarcq, C. Audoin, C. Salomon, European Physical Journal. D, 3, 201 (1998)

2. S. Bize, P. Laurent, M. Abgrall, H. Marion, I. Maksimovic, L. Cacciapuoti, J. Grnert, C. Vian, F. Pereira dos Santos, P. Rosenbusch, P.

Lemonde, G. Santarelli, P. Wolf, A. Clairon, A. Luiten, M. Tobar and C. Salomon, J.Phys. B: At.Mol.Opt.Phys., 38, S449-S468 (2005)

3. Ph. Laurent, M. Abgrall, Ch. Jentsch, P. Lemonde, G. Santarelli, A. Clairon, I. Maksimovic, S. Bize, Ch. Salomon, D. Blonde, J.F. Vega, O. Grosjean, F. Picard, M. Saccoccio, M. Chaubet, N. Ladiette, L. Guillet, I. Zenone, Ch. Delaroche, Ch. Sirmain, Appl. Phys. B 84, 683–690 (2006)

4. Luigi Cacciapuoti and Christophe Salomon, Space Clocks and Fundamental Tests: the ACES experiment, EPJ Special topics, 172, 57 (2009)

5. Laurent, D. Massonnet, L. Cacciapuoti, C. Salomon, The ACES /PHARAO space mission, Comptes-Rendus Acad. Sciences, Physique, Paris, 16, 540 (2015),

5. I. Moric, C. De Graeve, O. Grosjean, and Ph. Laurent, Rev. of Scient. Instruments 85, 075117 (2014)

6. Phillip Peterman, Kurt Gibble, Phillipe Laurent and Christophe Salomon, Metrologia 53 (2016) 899–907

- As for optical clocks, a review paper that should be cited is

Optical atomic clocks

Authors: N. Poli, C. W. Oates, P. Gill, G. M. Tino

pp. 555-624

Year 2013 - Issue 12

-> In the figures, error bars should be included.

-> The quality of some figures should be improved according to the journal standard.

-> At line 45, the sentence "for only several minutes" should be changed since in some cases those experiments last only for some seconds.

-> At line 47, references should be added (see above).

Reviewer #3 (Remarks to the Author):

Liu et al present a description of and results from their space-based atomic clock using cold atoms. Moving cold-atom experiments to space is a topic of great current interest, as is using this technology to improve applications like atomic clocks. As the first experiment of this nature, the Chinese effort is important and publication is well warranted.

Overall the paper is reasonably clear and covers the important points. Perhaps one exception is the issue of broadcasting the clock signal back to earth. This is not mentioned in the paper, but is certainly very important for any application. If a clock signal is available, it should be possible to compare to a stable terrestrial clock and obtain an Allan deviation plot such as Fig 8, but for the clock actually in operation in space. This would be considerably more interesting than the ground measurements presented. If the clock signal is not available, I think this should be explained and the impact on applications should be discussed.

There are however several smaller issues and clarifications that should be addressed.

1. In the introduction, the authors cite the best precision of a Cs fountain clock, which is of course much better than the performance seen here. For context, I think it would be useful to cite as well the stability of the current space-based GPS clocks. It would also be interesting to compare to the best commercially available clocks at this time.

2. In line 48, is PHARAO the name of the clock used in the ACES mission? The wording is a little confusing.

3. In line 70, "adiabatic cooling" is not explained. Eventually it is clarified in the methods section, but I think here the distinction between it and other laser cooling methods is not necessary. I would simply omit this line.

4. In lines 80-82, the description of the magnetic field coils is confusing. In particular the "C-field coil" is not defined. I suggest changing the wording to "The magnetic field in the interrogation region is stabilized to better than 5 nT using a servo loop and an electromagnet coil. This compensates for field variations arising from the motion of the spacecraft in orbit." Does the gradient field from the MOT remain on during the measurement?

5. The wording at lines 84-86 is confusing. I suggest "The MOT is formed using a pair of anti-Helmholtz coils and two laser beams with independently controlled frequencies. Each of the beams is folded in order to generate the multiple required trapping beams, and the geometry is designed so that a frequency shift between the beams forms a moving molasses that launches the atoms towards the interrogation region."

6. In line 99, I don't think that the CAC truly requires two cavities, since in principle a single cavity could be used with stationary atoms. I would replace "requires" with "uses."

7. In the discussion in lines 106-111, it would be interesting to report the total fraction of time since launch that the experiment has been operational.
8. I think section 2 should include additional quantitative details about the apparatus including: the number of atoms collected in the MOT, the power of the laser beams used, the range of accessible launch velocities, and the cycle time (repetition rate) of the measurements.
9. In Figure 2, how is the flight time defined? (I guess the time since the $\pi/2$ pulse?) I think it would be worth noting in the caption that this is the signal obtained for a microwave frequency such that most of the atoms end up in the $F=2$ state.
10. In Figure 3, how are the plotted values determined from the time of flight signals in Fig 2?
11. In line 139, the authors should explain why lower velocities lead to lower signal to noise. (I assume that the atoms spread out more during the longer transit, so that fewer of them are detected.) It would be useful here to describe or give a plot of the detected atom number as a function of launch velocity.
12. In Figure 4, what causes the coherence to decay at larger power? For part (b), what power level is used? (I guess -66 dBm or so.)
13. The meaning of the time scale in Figure 5 is not perfectly clear. The servo was locked at time $t = 0$?
14. At lines 170-171, it isn't entirely clear if the field compensation loop is open or closed. If it is close-loop, what does it mean that the parameters of the loop are specified by the real magnetic field? Also, how was the 4.5 nT value determined? It doesn't sound like there is a magnetometer in the actual interaction region.
15. In line 176, "circle" should probably be "orbit."
16. In Figure 7, the two figures seem to be swapped in relation to how they are described in the text. If I understand right, the situation illustrated in (a) is the case where the data are rejected. How are such events identified however?
17. The authors note a few situations where a measurement will be abandoned. In operations to date, what fraction of runs have been rejected in this way?
18. The discussion in lines 202-206 should I think be moved earlier to the paragraph at 137.
19. Is the clock period mentioned in line 207 the repetition rate? This should be provided earlier and the limits on faster operation discussed. Also in 207 the τ variable should be explained.
20. Figure 8 shows results obtained in ground operation. How was the experiment run on the ground, since gravity would deflect the atoms trajectory?
21. At lines 314, what are typical operating temperatures for the Rb source?
22. In Ext Data Fig 3, the actual beam path of the MOT beams is unclear; an extra panel illustrating this would be very helpful.

23. In the paragraph at 335, what is the waist of the detection beams? What is their power? What is the distance between them? Ultimately, what is the state detection accuracy? In other words, if no microwaves are applied and only $F=1$ atoms detected, how many $F=2$ will be erroneously recorded?

24. At line 364, I don't understand the relation between the 2 nT reported here and the 4.5 nT value mentioned previously.

25. In the laser description near 388, how much power do the lasers produce, and how much total is delivered to the experiment?

26. There are some confusing issues in the discussion of Method Section 2.2. First, how is Gaussian filtering used here? The detected signal depends on the size of the atom cloud, the size of the detection beams, and the light collection optics. This seems more complicated than the statement at lines 458-459 suggests. This all goes into determine the detection time Δt , but it is not clear how the authors handle this.

I don't understand the factor of E_{Rb} in Eq. (2). I think it should not be there. At line 466, I think γ should have units of s^{-1} .

27. In the discussion of the atom temperature measurement, it sounds like the authors assume that the atom cloud is very large compared to the detection beams. Is that accurate? Also, the actual TOF signal in Fig 2 does not appear to be a very accurate Gaussian, since it is noticeably asymmetric. Was it fit to a Gaussian anyway?

Response to Reviewers

Reviewer #1:

The authors present the first cold-atom experiment conducted in space, which is an amazing technical achievement. There has been a lot of work to try to send an atomic clock based on laser-cooled atoms into space over the past 20 years. There are several motivations for doing this. For one, under microgravity the atoms can be interrogated for long periods. Another reason is that the high speeds and microgravity environment make the system well suited for precision tests of relativistic effects. A third motivation is that cold-atom systems can potentially improve on the performance of current space based atomic clocks for GNSS systems.

Efforts to launch cold-atom clocks into space include a program in the US called PARCS, which was ended after the 2003 space shuttle disaster for lack of a delivery vehicle to the ISS. There has also been the ACES project in Europe, which continues work to launch a cold-atom atomic clock into space, but which was eclipsed by this Chinese space clock presented here by Liang and colleagues. The work presented by Liu and colleagues is made even more impressive by the detail that the Chinese clock program presented here did not start until 2011 – over ten years later than the start of the ACES program.

Because of the huge impact of this work, I highly recommend that the work be published in Nature Communications. However, I have quite a few grammar corrections and suggestions to recommend before publication. I also have a few suggestions for clarifications that should improve the technical content of the paper as well. I list these corrections below.

Response: We really appreciate the reviewer for the careful reading and a lot of comments on our work, which are really helpful for us to improve our manuscript. We have carefully checked and revised, and the details are as follow:

Detailed suggestions:

1. I made several editing suggestions for the manuscripts grammar directly onto the submitted PDF. I recommend that the authors carefully consider all suggestions to improve the readability of the manuscript – especially to the general audience that will read the paper.

Response 1 We have requested an English-professional editing services to improve our manuscript's grammars and we have made necessary corrections.

2. I would change the title to something like “In-orbit operation of an atomic clock based on laser-cooled 87Rb atoms”

Response 2 We have revised the title of our manuscript as the reviewer suggested.

3. Line 16: The abstract emphasizes that the width of the fringes scales linearly with launch velocity, but I don't think this is such an important detail that it should be emphasized in the abstract. Beam

clocks on the surface of the earth with the atom propagation direction perpendicular to gravity will also exhibit this behavior. I would remove this sentence from the abstract.

Response 3 Actually, compared with operation of our cold atom clock on the ground, here the “linearly” means operation in microgravity. We agree with the reviewer, here it is not worth to emphasize, and we have deleted this sentence in abstract.

4. Line 67: I would clarify the statement about the reduced density shift in the Rb clock and why that can improve performance. In my opinion, the smaller shift allows for relaxed control of the Rb density. So instead of saying “better performance can be expected”, I would recommend saying something like “better long-term performance can be expected with relaxed control required over the atomic density.”

Response 4 We agree with the reviewer and have changed the sentence as the reviewer suggested in line 67 of the revised manuscript.

5. On line 74 and then throughout the paper, I would change “physical package” to “physics package”.

Response 5 We agree with the reviewer and we have replaced the “physical package” with “physics package” throughout our manuscript.

6. Equation 2 should be an equality not a proportionality.

Response 6 We agree with the reviewer that the equation 2 should not be a proportionality. The linewidth of the central Ramsey fringes is $\Delta = v/2D$ under the conditions $\Delta\omega \ll \Omega$ and $T \gg \tau$. $\Delta\omega$ is the microwave frequency detuning to the resonance, and Ω is the Rabi frequency.

We have revised our manuscript in line 102 as “In two subsequent interaction of duration τ with the interrogating field separated by a free evolution time T , the Ramsey interrogation of a space CAC uses two cavities separated by a distance D

$$\Delta = v/2D \quad (T \gg \tau, \Delta\omega \ll \Omega) \quad (2),$$

where $\Delta\omega$ is the microwave frequency detuning to the resonance, and Ω the Rabi frequency.”

7. The sentence starting on line 110 sounds simplistic and I would change it to say something like: “The continuous microgravity time in orbit has allowed us to test the space CAC over a period of XX months.”

Response 7 We agree with the reviewer, and we have changed the sentence to “Under the almost continuous microgravity, our space CAC has already been tested for over fifteen months in orbit.” as the reviewer suggested in line 117 of our revised manuscript.

8. The two sentences beginning on line 126 are strange. I would suggest changing them to something like this: “The injected microwave power is optimized by measuring the transition probability described in Eq. (3). When the power is such that each passage through the microwave cavity applies a $\pi/2$ pulse, all of the population flips from ...”

Response 8 We agree with the reviewer, and have revised our manuscript accordingly (line 129-131).

9. The sentence beginning on line 129 is not true for all clocks, for example beam clocks with the beam path orthogonal to gravity. So I would change the beginning of the sentence to say “Unlike atomic fountains under gravity on the ground, ...”

Response 9 We agree with the reviewer, and have added the sentence “Unlike atomic fountains under gravity on the ground” in line 131 of the revised manuscript.

10. Do you know why there is a slight offset between the red line and the data points in Fig. 3? If so it would be good to comment about that bias.

Response 10 The measured linewidth is influenced by the size and temperature of the atomic cloud, the magnetic field distribution inside the Ramsey cavity, as well as the actual microgravity level in orbit. We think the small offset may come from these contributions.

11. The sentence beginning on line 143 needs clarification for a few reasons. Also, which cavity are you scanning the power in? The state selection or Ramsey cavity? If it's the Ramsey cavity, don't you want to apply $\pi/2$ pulses? And are you turning the microwaves off during the second passage through the cavity?

Response 11 We scan the power of the injected microwave in the Ramsey cavity. The microwave is not off during the second passage through the cavity.

Actually, we scan the microwave power injected into the cavity in a Ramsey Scheme (two microwave pulse separated by a free flight), during the free flight the microwave is turned off. The microwave power for the each required $\pi/2$ transition is determined by the first peak of the Rabi fringes.

We have revised the text in line 147-148 of the revised manuscript as “Fig. 4 (A) presents a plot of the Rabi oscillation versus microwave power at resonance frequency to determine the power required for $\pi/2$ transition in each interaction zone”

12. I'm not sure if I agree with the conclusion in the sentence beginning on line 144. At higher microwave power, you expect the contrast to go down because of inhomogeneities in the microwave amplitude across the atom cloud. I would argue that the reduced contrast is not from sparse measurements. But how is the theoretical curve calculated and what is assumed? It seems like there is another cause of decay that is not accounted for, or that the inhomogeneities in the cavity are larger than expected.

Response 12 We agree with the reviewer on the explanation of the contrast at the higher microwave power. Actually, besides the inhomogeneities in the microwave amplitude, the velocity distribution of the atomic cloud is another important factor to affect the contrast at higher microwave power. In our calculation, the temperature of the atomic cloud is set as 3.3 μK which is an averaged result of the temperature measurements. The temperature in this scanning is probably little higher than that used in the calculation which caused a faster decay at higher microwave power. We use the first peak to determine the microwave power for a required $\pi/2$ transition, which is affected very slightly by the two factors mentioned above. Therefore, other peaks are not important in our case.

We have added these discussions in line 151 of our revised manuscript as “mainly due to the velocity distribution of the cold atomic cloud and the inhomogeneities in the microwave amplitude. We focused on the first peak...”

13. I think the discussion of figures 7A and 7B has the cases backwards in the text. Please check. Maybe you should change the text to say something like this: “When the spikes are actually on the TOF signal as shown in Fig. 7A, the data is thrown out. When the spikes are near the TOF signal but well separated from the signal, a window function is used to “filter out” the TOF signal from the contaminated signal.”

Response 13 We have checked the discussion about figures 7A and 7B according to the reviewer’s suggestion and have made correction in the revised manuscript.

14. The discussion from lines 202 through the beginning of 206 is redundant with the earlier discussion of the SNR versus atom launch velocity.

Response 14 We agree with the reviewer and we have omitted the sentence in our revised manuscript.

15. I don’t think it will be clear to all readers why you can’t show an Allan Deviation for the clock performance in flight. I would suggest adding a sentence starting on line 208 to say something like this: “There is no second clock to compare the CAC to aboard the spacecraft, so we have no reference with which to evaluate the clock stability in orbit.”

Response 15 We really appreciate the reviewer giving the suggestion to improve our manuscript. We have revised the sentence in line 212 of the revised manuscript as “There is no second clock on-board and also no frequency dissemination link to the ground, so we have no reference with which to evaluate the clock stability in orbit.”

16. The estimated stability in orbit is actually a lot better than what you showed on the ground. I would emphasize that by saying something like: “...the estimated instability in orbit is more than a factor of six lower than on earth, which we attribute to the microgravity environment...”

Response 16 We really appreciate the reviewer giving the suggestion to improve our manuscript. We have revised the sentence in line 218 of the revised manuscript as “The estimated instability

in orbit is more than a factor of six lower than on Earth, which we attribute to the microgravity environment”.

17. Caption of Fig. 8: Please specify what the launch velocity was for the ground based measurement and what the orientation of the CAC was with respect to gravity.

We have revised the caption of Fig.8 by adding a sentence “The cold atomic cloud is launched downward with a velocity of 1 m s^{-1} on the ground.”

Reviewer #2

The paper by Liang Liu et al. reports on the first demonstration of the operation of a cold atom clock in microgravity. Even if the reported results can be considered still as proof-of-principle, this is in my opinion an important milestone towards experiments with cold atoms in space and therefore the paper is worth being published.

I would suggest however that the author consider the following comments in order to submit an improved version.

Response: We really appreciate the reviewer for carefully reading the manuscript and providing helpful comments which help us improve the manuscript.

-> English should be checked and improved.

Response: We have requested an English-professional editing services to improve our manuscript's grammars and made necessary changes.

-> Since the work reported in this paper is similar to the ACES/Pharao CNES planned mission, ACES should be cited more clearly and the results of this work should be compared with the one expected from ACES.

Response: We have added more information about the ACES mission and the PHARAO clock in line 49-54 of the revised manuscript as following:

“..... which consists of a cesium CAC called PHARAO, a hydrogen maser as well as a package for frequency comparison and distribution, aims to search for drifts in fundamental constants and measure the gravitational red shift with improved precision¹⁹⁻²⁵. The PHARAO clock is expected to operate in space with a frequency stability of $1 \times 10^{-13} \tau^{-1/2}$ and an accuracy below 3×10^{-16} .”

-> Some references are missing and some could be omitted or replaced;

- Ref.s 3 and 4 could be omitted and replaced with relevant references on the characterization of the best fountain clocks.

- Reference to planned future experiments with atomic sensors in space should be included as, for example:

Atom interferometers and optical atomic clocks: New quantum sensors for fundamental physics experiments in space

Nuclear Physics B - Proceedings Suppl. Volume 166, Pages 159-165 (2007)

- References to ACES/Pharao should be included: a list from which the authors may choose is:

1. P. Laurent, P. Lemonde, E. Simon, G. Santarelli, A. Clairon, P. Petit, N. Dimarcq, C. Audoin, C. Salomon, European Physical Journal. D, 3, 201 (1998)
2. S. Bize, P. Laurent, M. Abgrall, H. Marion, I. Maksimovic, L. Cacciapuoti, J. Grnert, C. Vian, F. Pereira dos Santos, P. Rosenbusch, P. Lemonde, G. Santarelli, P. Wolf, A. Clairon, A. Luiten, M. Tobar and C. Salomon, J.Phys. B: At.Mol.Opt.Phys., 38, S449-S468 (2005)
3. Ph. Laurent, M. Abgrall, Ch. Jentsch, P. Lemonde, G. Santarelli, A. Clairon, I. Maksimovic, S. Bize, Ch. Salomon, D. Blonde, J.F. Vega, O. Grosjean, F. Picard, M. Saccoccio, M. Chaubet, N. Ladiette, L. Guillet, I. Zenone, Ch. Delaroche, Ch. Sirmain, Appl. Phys. B 84, 683–690 (2006)
4. Luigi Cacciapuoti and Christophe Salomon, Space Clocks and Fundamental Tests: the ACES experiment, EPJ Special topics, 172, 57 (2009)
5. Laurent, D. Massonnet, L. Cacciapuoti, C. Salomon, The ACES /PHARAO space mission, Comptes-Rendus Acad. Sciences, Physique, Paris, 16, 540 (2015),
5. I. Moric, C. De Graeve, O. Grosjean, and Ph. Laurent, Rev. of Scient. Instruments 85, 075117 (2014)
6. Phillip Peterman, Kurt Gibble, Phillipe Laurent and Christophe Salomon, Metrologia 53 (2016) 899–907

- As for optical clocks, a review paper that should be cited is

Optical atomic clocks

Authors: N. Poli, C. W. Oates, P. Gill, G. M. Tino

pp. 555-624

Year 2013 - Issue 12

Response: We really appreciate the reviewer for giving many helpful references and we have updated the reference list accordingly in our revised manuscript.

-> In the figures, error bars should be included.

Response: In Figure 3(E), we have added error bars for 0.6 m s^{-1} , 1.0 m s^{-1} , 2.0 m s^{-1} , 3.0 m s^{-1} and 4.0 m s^{-1} . For each of other velocities, only one measurement was performed due to the experiment time limited by the management of the Tiangong-2.

-> The quality of some figures should be improved according to the journal standard.

Response: We have improved the quality of some figures according to the journal standard in the revised manuscript.

-> At line 45, the sentence "for only several minutes" should be changed since in some cases those experiments last only for some seconds.

Response: In line 44-46 of the revised manuscript, we have changed the sentence to “These methods provide a microgravity environment ranging from several seconds (drop tower, parabolic flight) to several minutes (sounding rocket)”

-> At line 47, references should be added (see above).

Response: We have added the references.

Reviewer #3

Liu et al present a description of and results from their space-based atomic clock using cold atoms. Moving cold-atom experiments to space is a topic of great current interest, as is using this technology to improve applications like atomic clocks. As the first experiment of this nature, the Chinese effort is important and publication is well warranted.

Overall the paper is reasonably clear and covers the important points. Perhaps one exception is the issue of broadcasting the clock signal back to earth. This is not mentioned in the paper, but is certainly very important for any application. If a clock signal is available, it should be possible to compare to a stable terrestrial clock and obtain an Allan deviation plot such as Fig 8, but for the clock actually in operation in space. This would be considerably more interesting than the ground measurements presented. If the clock signal is not available, I think this should be explained and the impact on applications should be discussed.

There are however several smaller issues and clarifications that should be addressed.

Response: We highly appreciate the reviewer for carefully reading the manuscript and providing helpful comments which help us improve the manuscript.

As the reviewer commented, a high precision time and frequency link between the spacecraft and the earth is important for many applications. However, there is no such a link between the Tiangong-2 and the earth due to the restricted on-board resources. The link will be established in our next space mission. To clarify this situation, we have added such a sentence in line 212-214 the revised manuscript: “There is no second clock on-board and also no frequency dissemination link to the ground, so we have no reference with which to evaluate the clock stability in orbit.”

There are also other helpful suggestions and issues needed to clarify. We have given necessary emendations and explanations in the manuscript as the list below according to the reviewer’s comments.

1. In the introduction, the authors cite the best precision of a Cs fountain clock, which is of course much better than the performance seen here. For context, I think it would be useful to cite as well the stability of the current space-based GPS clocks. It would also be interesting to compare to the best commercially available clocks at this time.

Response 1 This is a helpful suggestion. We have added a sentence in line 36: “Currently, the best performing space atomic clocks used in the GNSS systems are those at frequency stability of a few parts in 10^{15} per day.....”

2. In line 48, is PHARAO the name of the clock used in the ACES mission? The wording is a little confusing.

Response 2 Yes, the PHARO is the name of the cold clock used in the ACES mission. To avoid the confusion, we have revised the sentence in line 49-50 as: “For example, the ACES mission, which consists of a CAC called PHARAO”.

3. In line 70, “adiabatic cooling” is not explained. Eventually it is clarified in the methods section, but I think here the distinction between it and other laser cooling methods is not necessary. I would simply omit this line.

Response 3 We agree this comment and have omitted that line.

4. In lines 80-82, the description of the magnetic field coils is confusing. In particular the “C-field coil” is not defined. I suggest changing the wording to “The magnetic field in the interrogation region is stabilized to better than 5 nT using a servo loop and an electromagnet coil. This compensates for field variations arising from the motion of the spacecraft in orbit.” Does the gradient field from the MOT remain on during the measurement?

Response 4 We have revised the sentence to “The magnetic field in the interrogation region is stabilized to better than 5 nT using a servo loop and a magnetic field coil, which compensate for field variations arising from the motion of the spacecraft in orbit” in line 83-85 of the revised manuscript.

The gradient field from the MOT is turned off after launching the cold atoms until the beginning of the next clock cycle.

5. The wording at lines 84-86 is confusing. I suggest “The MOT is formed using a pair of anti-Helmholtz coils and two laser beams with independently controlled frequencies. Each of the beams is folded in order to generate the multiple required trapping beams, and the geometry is designed so that a frequency shift between the beams forms a moving molasses that launches the atoms towards the interrogation region.”

Response 5 We have revised our manuscript according to the reviewer’s suggestion in line 87-91 as “The MOT is formed using a pair of anti-Helmholtz coils and two laser beams with independently controlled frequencies. Each beam is folded to create the multiple trapping beams required.....the geometry is designed so that a frequency shift between the beams forms a moving molasses that launches atoms towards the interrogation region with a velocity”.

6. In line 99, I don’t think that the CAC truly requires two cavities, since in principle a single cavity could be used with stationary atoms. I would replace “requires” with “uses.”

Response 6 We have replaced “requires” with “uses” in our revised manuscript.

7. In the discussion in lines 106-111, it would be interesting to report the total fraction of time since launch that the experiment has been operational.

Response 7 Up to now, the total in-orbit time is about 18 months, among which the operational time has exceeded fifteen months

We have added this information in line 114-118 of our revised manuscript as “It was launched ...Under the almost continuous microgravity, our space CAC has already been tested for over fifteen months in orbit.”

8. I think section 2 should include additional quantitative details about the apparatus including: the number of atoms collected in the MOT, the power of the laser beams used, the range of accessible launch velocities, and the cycle time (repetition rate) of the measurements.

Response 8 The number of atoms collected in the MOT is about 5×10^7 with the power of each laser beam about 3.8 mW/cm^2 . The launch velocities range from 0.6 to 6.0 m/s in orbit.

The cycle time of the measurement is about 2.0 s which we have mentioned in line 212 of the revised manuscript.

We have add the quantitative details mentioned above in section “Results” of the revised manuscript as following:

- We have added a sentence in line 89 as “The power of each trapping beam is about 3.8 mW cm^{-2} and the geometry...”
- We have added a sentence in line 94 as “With this process, about 5×10^7 cold atoms are launched from the MOT zone when the loading time is set to 1.0 s.”
- We have added a sentence in line 94 as “The velocity ranging from 0.6 to 6.0 m s^{-1} is accessible.”

9. In Figure 2, how is the flight time defined? (I guess the time since the $\pi/2$ pulse?) I think it would be worth noting in the caption that this is the signal obtained for a microwave frequency such that most of the atoms end up in the $F=2$ state.

Response 9 In Figure 2, the flight time begins when the cold atomic cloud is launched. And the signal in Figure 2 is obtained after a free flight from the MOT zone to the detection zone and the microwave is off in the whole process.

The TOF signal in Figure 2 is obtained without microwave interaction.

10. In Figure 3, how are the plotted values determined from the time of flight signals in Fig 2?

Response 10 The TOF signals in Fig 2 are obtained without microwave interaction. The plot values in Fig 3 are calculated from the TOF signals with microwave interaction using the equation

$$p = \frac{N_2}{N_1 + N_2}$$
, where N_1 and N_2 are the measured number of the cold atoms in the state $|F = 1\rangle$ and $|F = 2\rangle$ respectively.

11. In line 139, the authors should explain why lower velocities lead to lower signal to noise. (I assume that the atoms spread out more during the longer transit, so that fewer of them are detected.) It would be useful here to describe or give a plot of the detected atom number as a function of launch velocity.

Response 11 Neglecting the technical noise, the SNR is proportional to $\sqrt{N_{det}}$, where N_{det} is number of the detected atoms. There are two major loss mechanisms related to the launch velocities during the flight of the cold atomic cloud: the collision with the background gas and the expansion of the cloud. These kinds of loss are time-related. Lower velocity induces larger loss of cold atoms which lead to a lower SNR.

12. In Figure 4, what causes the coherence to decay at larger power? For part (b), what power level is used? (I guess -66 dBm or so.)

Response 12 The velocity distribution of the atomic cloud and the inhomogeneities of the microwave amplitude in cavity lead the decay of the coherence at higher microwave power.

As the reviewer's comment, the power used for Figure 4(b) is about -66 dBm, to be accurate is -66.5 dBm. We have added this information in line 152-154 of the revised manuscript as "We focused on the first peak, from which we got the microwave power for the required $\pi/2$ transitions is about -66.5 dBm; the corresponding Ramsey fringes are given in Fig. 4 (B)."

13. The meaning of the time scale in Figure 5 is not perfectly clear. The servo was locked at time $t = 0$?

Response 13 We have revised our manuscript by adding this information in line 167 as "The servo loop is activated at time $t=0$."

14. At lines 170-171, it isn't entirely clear if the field compensation loop is open or closed. If it is close-loop, what does it mean that the parameters of the loop are specified by the real magnetic field? Also, how was the 4.5 nT value determined? It doesn't sound like there is a magnetometer in the actual interaction region.

Response 14 In the compensation loop of the CAC, a monitoring magnetometer is set in two layer shields near the detection zone, the monitoring values of magnetic field is used to as the feedback. In orbit operation, we try different parameters to minimize the magnetic field variations along the trajectory of the cold atoms inside the interrogation zone which are measured using a magnetically sensitive transition. Using the optimized parameter we get the 4.5 nT.

We use a magnetically sensitive transition from $|F = 1, m_F = -1\rangle$ to $|F = 2, m_F = -1\rangle$ of the ^{87}Rb to the magnetic field along the trajectory of the cold atoms inside the interrogation zone by scanning the frequency of the microwave. We therefore get the resonance frequency of the magnetically sensitive transition. The shift of the magnetically sensitive transition frequency is determined by $\frac{\Delta\nu}{\Delta B} \approx 14 \text{ Hz/nT}$.

15. In line 176, "circle" should probably be "orbit."

Response 15 We agree with the reviewer and we have replaced the word “circle” with “orbit”.

16. In Figure 7, the two figures seem to be swapped in relation to how they are described in the text. If I understand right, the situation illustrated in (a) is the case where the data are rejected. How are such events identified however?

Response 16 The discussions of the two figures are reversed and we have revised our manuscript by correcting the discussion of this two figures in line 192-195 of the revised manuscript as “When the spikes are actually on the TOF signal (Fig. 7A), the data are discarded. When the spikes are near the TOF signal but well separated from the signal (Fig. 7B), a window function is used to filter out the TOF signal from the contaminated signal”.

During the data processing, a peak count function combining with a differential method is used to identify the two events.

17. The authors note a few situations where a measurement will be abandoned. In operations to date, what fraction of runs have been rejected in this way?

Response 17 The spikes only occurs when the aircraft flies over the SAA (South Atlantic Anomaly) region. The data obtained out of the SAA region does not need to be abandoned, while in the SAA region, the spikes just accidentally appear actually on the TOF that need to be abandoned and most of the spikes appear near the TOF signal as Figure 7(b) that does not need to be abandoned. Up to now, the fraction of runs have been rejected is about 0.05%.

18. The discussion in lines 202-206 should I think be moved earlier to the paragraph at 137.

Response 18 In our initial manuscript, the discussion from lines 202 through the beginning of 206 is redundant with the earlier discussion of the SNR versus atom launch velocity in line 137. So we have omitted this sentence in our revised manuscript.

19. Is the clock period mentioned in line 207 the repetition rate? This should be provided earlier and the limits on faster operation discussed. Also in 207 the τ variable should be explained.

Response 19 In line 207 of the initial manuscript, the clock period is the repetition rate which is limited by the loading time of the cold atoms and the flight time of the cold atomic cloud. The choice of the clock period is a trade of the cold atoms number, the linewidth of the Ramsey and the dead time.

We have explained the τ variable in line 211-212 of the revised manuscript as “a short-term frequency stability at an averaging time τ is close to...”

20. Figure 8 shows results obtained in ground operation. How was the experiment run on the ground, since gravity would deflect the atoms trajectory?

Response 20 We have added the information in the caption of Figure 8: “The cold atomic cloud is launched downward with a velocity of 1 m s^{-1} on the ground”

21. At lines 314, what are typical operating temperatures for the Rb source?

Response 21 The temperature of Rb source was controlled using a server loop. The typical set-point is 35 degree centigrade.

22. In Ext Data Fig 3, the actual beam path of the MOT beams is unclear; an extra panel illustrating this would be very helpful.

Response 22 We have added the scheme of the beam path in Ext Data Fig 3 as following.

23. In the paragraph at 335, what is the waist of the detection beams? What is their power? What is the distance between them? Ultimately, what is the state detection accuracy? In other words, if no microwaves are applied and only F=1 atoms detected, how many F=2 will be erroneously recorded?

Response 23 As described in our manuscript, there are four laser beams in the detection zone. The waist of each probing beam is about 7 mm. while the repumping and pushing beam are about 2 mm. The distance between two probing beams is about 30 mm. There is about 8% state detection error in this system due to the atoms in state F=2 not being pushed away completely by the pushing beam. This will cause a contrast reduction of the Ramsey fringes and lead to a degradation of frequency stability.

24. At line 364, I don't understand the relation between the 2 nT reported here and the 4.5 nT value mentioned previously.

Response 24 At line 364 of the initial manuscript, the result of 2 nT represents the inhomogeneity of the magnetic field inside the three-layer field with static geomagnetic field on the ground. While 4.5 nT measured in orbit is the time variations of the magnetic field inside the interrogation zone.

25. In the laser description near 388, how much power do the lasers produce, and how much total is delivered to the experiment?

Response 25 Response: There are one cooling laser and one repumping laser in clock operation. The cooling laser outputs 110 mW and about 36 mW is delivered to the physics package. The main loss is from the AOMs, the optical isolators and the fiber-coupling. The repumping laser outputs 100mW and about 6 mW is intended to deliver to the physics package.

26. There are some confusing issues in the discussion of Method Section 2.2. First, how is Gaussian filtering used here? The detected signal depends on the size of the atom cloud, the size of the detection beams, and the light collection optics. This seems more complicated than the statement at lines 458-459 suggests. This all goes into determine the detection time Δt , but it is not clear how the authors handle this.

I don't understand the factor of E_{Rb} in Eq. (2). I think it should not be there. At line 466, I think γ should have units of s^{-1} .

Response 26 In Method Section 2.2 of the initial manuscript, the atoms number are calculated using the equation,

$$N = \frac{\int U_{PD}(t)dt}{\gamma_p \cdot \Delta t \cdot c \cdot \rho_{PD} \cdot G \cdot E_{Rb}}$$

The procedure of the atom number is determined as follow:

The cold atoms are excited by the probing beam to emit fluorescence. For a single cold atom, the power of emitted fluorescence is $\gamma_p \cdot E_{Rb}$. Assuming all the cold atoms have a same velocity passing through the probing beam, they have the same passing time Δt . Therefore, $\gamma_p \cdot E_{Rb} \cdot \Delta t$ gives the energy of fluorescence emitted by a single atom.

Only part of the fluorescence is collected by a photodiode. The collection efficiency is c . The quantum efficiency of the photodiode is ρ_{PD} . The photocurrent is converted into voltage TOF signal U_{PD} by an amplifier with a gain of G . Integration of the TOF signal divided by $c \cdot \rho_{PD} \cdot G$ gives the energy of fluorescence emitted by all atoms.

Then the number of the atom is obtained by the above equation

In the equation, γ_p is the scattering rate of atoms, which is related to the laser power and frequency detuning. We have added the unit of s^{-1} for γ_p in our manuscript.

In data processing, we use a Gaussian fitting to filter out the technical noise, such as detection electronics noise, noise due to stray light and thermal beam and laser noise.

27. In the discussion of the atom temperature measurement, it sounds like the authors assume that the atom cloud is very large compared to the detection beams. Is that accurate? Also, the actual TOF signal in Fig 2 does not appear to be a very accurate Gaussian, since it is noticeably asymmetric. Was it fit to a Gaussian anyway?

Response 27 It is true that the thickness of the detection beam is ignored in the discussion of the atom temperature measurement. Actually, this assumption will introduce errors because the TOF signal is the convolution of the atoms cloud and the detection beams during the cloud passing the beams. More accurate temperature measurement can be obtained by using the deconvolution method. In our preview works we find the temperature measurement errors depend on the value of

atoms temperature in a similar detection system. At a 3 μk temperature of the atoms cloud, the error is about 0.4 μk . We made an approximate estimation here.

Yes, we still use Gaussian fitting in the TOF data procession even though there is a tiny asymmetry in the TOF signals.

Reviewer #1 (Remarks to the Author):

The revised manuscript does a great job at addressing most of the reviewers' concerns. I would recommend it for publication after a few more minor edits and corrections are made. I recommend a few changes below, and I have also sent a PDF with some additional grammatical edits that I recommend.

On page 2, line 44, consider also citing this paper:

G. Stern, B. Battelier, R. Geiger, G. Varoquaux, A. Villing, F. Moron, et al., "Light-pulse atom interferometry in microgravity," *The European Physical Journal D*, vol. 53, pp. 353-357, 2009.

On Page 2, line 68, consider also citing this paper:

Y. Sortais et al., "Cold Collision Frequency Shifts in a 87Rb Atomic Fountain," *Physical Review Letters*, vol. 85, pp. 3117-3120, 2000.

On Page 6, line 150 – To describe Fig. 4 as “Rabi Oscillations” seems just slightly off, since usually Rabi oscillations are versus time at a fixed microwave power. So I would change the description slightly and call them “population oscillations”.

Line 267 in the methods section – I would change the text to “with a stability of +/- 0.01 °C.” The use of the word “accuracy” implies that it is calibrated, but earlier in the sentence it says “... about 25 C.” The use of the word “accuracy” implies that it is calibrated, but earlier in the sentence it says “... about 25 °C”, which is contradictory.

Line 269 in the methods section – The text says “Because the CAC does not incur the problems as for a thermal atom beam,” but I don't understand what is meant by this. Usually probe beams are perpendicular to the atom trajectories even when thermal beams are used. So you might consider removing that statement or explaining better what is meant by it.

Line 274 in the methods section – I think that the states for the repumping are mislabeled. Please check.

Methods section, page 10 – The labeling of the laser diodes at the bottom of page 10 is confusing. Please check the labels.

Please rewrite the Extended Data Fig. 7 caption. I made some suggestions.

Reviewer #2 (Remarks to the Author):

The authors have considerably improved the manuscript taking into account the reviewers' comments and suggestions. In my opinion, this paper can be published in its present form without delay.

Reviewer #3 (Remarks to the Author):

Liu et al have effectively addressed the comments of myself and I think the other reviewers. I do still have a few concerns, however, mainly in places where the authors answered a question in their response but I think it would be beneficial to include that information in the manuscript itself.

Line 53: The authors must define tau and provide units for the stability figure. Alternatively, they might cite "a frequency stability of 1×10^{-13} at an averaging time of 1 s." Note that tau is used differently here than in line 103.

Line 89: The authors wrote power when they meant intensity (or irradiance).

Figures 2 and 3: This is one place where I remain confused. What exactly are the two signals in Fig 2? I understand that the atoms pass through two detection beams, but there is only one fluorescence detector, correct? It would seem then that only one signal is available, and I don't understand how the red and black curves were resolved. (My initial impression was that it involved different types of state preparation, but from the authors' response that seems not to be the case.) Similarly, it isn't very clear how the N1 and N2 populations are actually determined. The methods section discusses a Gaussian fit, but if there is only one fluorescence signal, then I guess it should appear as the sum of two Gaussians. Moreover, the F=1 trace appears rather non-Gaussian. To be clear, I have no concerns that the authors are doing their detection in a sensible way. I just don't feel it has been explained clearly enough (for me at any rate).

Line 146: I think the manuscript should include a brief explanation of why the lower velocity leads to lower SNR.

Line 180: I think the manuscript should include a brief explanation of how the 4.5 nT result was obtained.

Line 211: I think the manuscript should give the 0.05% rejection rate mentioned in the response.

Line 369: I think this would be clearer if it were stated that U is measured in volts.

Response to Reviewers

Reviewer #1:

The revised manuscript does a great job at addressing most of the reviewers' concerns. I would recommend it for publication after a few more minor edits and corrections are made. I recommend a few changes below, and I have also sent a PDF with some additional grammatical edits that I recommend.

Response: We really appreciate the reviewer for the careful reading and recommending on our manuscript. We have carefully checked the grammar and revised them according to the reviewer's suggestions. In addition, there are other changes or corrections as follow:

On page 2, line 44, consider also citing this paper:

G. Stern, B. Battelier, R. Geiger, G. Varoquaux, A. Villing, F. Moron, et al., "Light-pulse atom interferometry in microgravity," *The European Physical Journal D*, vol. 53, pp. 353-357, 2009.

On Page 2, line 68, consider also citing this paper: Y. Sortais et al., "Cold Collision Frequency Shifts in a 87 Rb Atomic Fountain," *Physical Review Letters*, vol. 85, pp. 3117-3120, 2000.

Response: We appreciate the reviewer for giving the two helpful references and we have updated the reference list accordingly in our revised manuscript.

On Page 6, line 150 – To describe Fig. 4 as “Rabi Oscillations” seems just slightly off, since usually Rabi oscillations are versus time at a fixed microwave power. So I would change the description slightly and call them “population oscillations”.

Response: We agree with the reviewer and we have replaced the “Rabi Oscillations” with “population oscillations” in our manuscript.

Line 267 in the methods section – I would change the text to “with a stability of +/- 0.01 °C.” The use of the word “accuracy” implies that it is calibrated, but earlier in the sentence it says “... about 25 C.” The use of the word “accuracy” implies that it is calibrated, but earlier in the sentence it says “... about 25 ° C”, which is contradictory.

Response: We agree with the reviewer and have replaced the “accuracy” with “stability” as the reviewer suggested in line 269.

Line 269 in the methods section – The text says “Because the CAC does not incur the problems as for a thermal atom beam,” but I don't understand what is meant by this. Usually probe beams are perpendicular to the atom trajectories even when thermal beams are used. So you might consider removing that statement or explaining better what is meant by it.

Response: We agree with the reviewer and removed that statement in line 270.

Line 274 in the methods section – I think that the states for the repumping are mislabeled. Please check. Methods section, page 10 – The labeling of the laser diodes at the bottom of page 10 is confusing. Please check the labels.

Response: It was mislabeled for the repump laser in line 275 and in line 313. We have revised them to “ $|5^2S_{1/2}, F = 1\rangle \rightarrow |5^2P_{3/2}, F = 2\rangle$ ” and “ $|5^2S_{1/2}, F = 1\rangle \rightarrow |5^2P_{3/2}, F = 1, 2\rangle$ ” respectively.

Please rewrite the Extended Data Fig. 7 caption. I made some suggestions.

Response: We have revised the coordinate axes and rewritten the caption of Extended Data Fig. 7 as “Axial variation of the magnetic field inside the three-layer shield under static geomagnetic condition. The zero position of distance is defined at the axial center of the interrogation cavity.”.

Reviewer #2 (Remarks to the Author):

The authors have considerably improved the manuscript taking into account the reviewers' comments and suggestions. In my opinion, this paper can be published in its present form without delay.

Response: We really appreciate the reviewer's many efforts on our manuscript.

Reviewer #3 (Remarks to the Author):

Liu et al have effectively addressed the comments of myself and I think the other reviewers. I do still have a few concerns, however, mainly in places where the authors answered a question in their response but I think it would be beneficial to include that information in the manuscript itself.

Response: We appreciate the reviewer for the efforts to improve the manuscript.

1. Line 53: The authors must define tau and provide units for the stability figure. Alternatively, they might cite “a frequency stability of 1×10^{-13} at an averaging time of 1 s.” Note that tau is used differently here than in line 103.

Response 1: We have added the definition of tau in line 54: (τ is average time in second). Additional, we have changed “ τ ” to “ Δt ” in line 104 and 109 respectively to distinguish the previous definition on tau.

2. Line 89: The authors wrote power when they meant intensity (or irradiance).

Response 2: We have replaced “power” with “intensity” in line 89.

3. Figures 2 and 3: This is one place where I remain confused. What exactly are the two signals in Fig 2? I understand that the atoms pass through two detection beams, but there is only one fluorescence

detector, correct? It would seem then that only one signal is available, and I don't understand how the red and black curves were resolved. (My initial impression was that it involved different types of state preparation, but from the authors' response that seems not to be the case.) Similarly, it isn't very clear how the N1 and N2 populations are actually determined. The methods section discusses a Gaussian fit, but if there is only one fluorescence signal, then I guess it should appear as the sum of two Gaussians. Moreover, the F=1 trace appears rather non-Gaussian. To be clear, I have no concerns that the authors are doing their detection in a sensible way. I just don't feel it has been explained clearly enough (for me at any rate).

Response 3: We have two photodetectors in our CAC, the detection process is shown as following: There are four beams in the detection zone. The first beam is circularly polarized and forms a standing wave tuned to the transition $|F = 2\rangle \rightarrow |F' = 3\rangle$ and a photodetector is used to detect the population in state $|F = 2\rangle$ (Fig.2 black curve). Tuned to the same transition as the first, the second beam is a travelling wave to push away the atoms in state $|F = 2\rangle$ after their detection. The third beam is tuned to the transition $|F = 1\rangle \rightarrow |F' = 2\rangle$ and pumps the atoms in state $|F = 1\rangle$ to state $|F = 2\rangle$. Finally, the fourth beam is set up like the first and another photodetector detects the remaining atoms which represent the population in state $|F = 1\rangle$ (Fig.2 red curve). This is a typical method used in atomic fountain clocks.

As to the determination of the N1 and N2 populations, we use the equation $N = \frac{\int U_{PD}(t)dt}{\gamma_P \cdot \Delta t \cdot c \cdot \rho_{PD} \cdot G \cdot E_{Rb}}$ as presented in methods section. Firstly, we fit the TOF signals of $|F = 1\rangle$ and $|F = 2\rangle$ respectively using the Gaussian shape to filter out the technical noise due to the detection process and yield the $U_{PD}(t)$ curves. Substituting these curves and other technical parameters into above equation, the N1 and N2 populations are obtained.

In Fig.2, the shape of F=1 trace looks non-Gaussian because the trace is much smaller compared to the F=2 trace and there exists an interfering tiny bump near the TOF signal. The bump comes from the unblocked stray light. In fact, the lineshape of the TOF signal in F=1 trace is Gaussian.

4. Line 146: I think the manuscript should include a brief explanation of why the lower velocity leads to lower SNR.

Response 4: We have included the explanation in line 146: "..., it also leads to a lower SNR mainly due to the loss of cold atoms originated from expansion of cold atom cloud...."

5. Line 180: I think the manuscript should include a brief explanation of how the 4.5 nT result was obtained.

Response 5: We have included the explanation in line 181: "..., The magnetic field along the trajectory of the cold atoms inside the interrogation zone is measured by exciting the magnetic-sensitive transition of rubidium²⁴ and a variation of 4.5 nT is obtained when the loop is active....". We have cited Ref. 24 for the magnetic-sensitive transition method.

6. Line 211: I think the manuscript should give the 0.05% rejection rate mentioned in the response.

Response 6: We have added this information in line 201: “..., the influence from discarded data (about 0.05% of the total data) to the clock operation is slight and can be ignored...”

7. Line 369: I think this would be clearer if it were stated that U is measured in volts.

Response 7: We have changed the sentence in line 372 as: “..., the TOF signal lineshape in volts, ...”

REVIEWERS' COMMENTS:

Reviewer #3 (Remarks to the Author):

The authors have addressed my concerns, and I recommend for publication. I am not sure why I became confused that there was only one detection photodiode, certainly the paper says there are two. With that clarification, the detection scheme makes sense. I might, however, suggest that the traces in Figure 2 be replaced with data showing comparable populations in each state. This is at the author's option.

Reviewer #3 (Remarks to the Author):

The authors have addressed my concerns, and I recommend for publication. I am not sure why I became confused that there was only one detection photodiode, certainly the paper says there are two. With that clarification, the detection scheme makes sense. I might, however, suggest that the traces in Figure 2 be replaced with data showing comparable populations in each state. This is at the author's option.

Response: We really appreciate the reviewer for the efforts to improve the manuscript. From our point of view, however, we prefer to keep Figure 2 unchanged. The figure in current way, we think, can better represent the performance of our detection system such as the dynamic range of the acquisition of TOF signals.